# What Are the Economic Arguments for Mandating LGBT+ Health Training for Healthcare Providers? An Economic Evaluation of the Impacts of LGBT+ Health Training on Cervical Screening

**DOI:** 10.3390/bs14030260

**Published:** 2024-03-21

**Authors:** Saima Bashir, William Whittaker, Catherine Meads

**Affiliations:** 1Manchester Centre for Health Economics, University of Manchester, Oxford Road, Manchester M13 9PL, UK; william.whittaker@manchester.ac.uk; 2Faculty of Health, Medicine and Social Care, Anglia Ruskin University, Chelmsford CM1 1SQ, UK; catherine.meads@aru.ac.uk

**Keywords:** equity, LGBTQ+ training, England, smear screening, cervical cancer

## Abstract

**Background**: Equitable access to healthcare is a priority of many healthcare systems, aiming to ensure access is driven by need and not minority groups such as those defined by sexual orientation. However, there are healthcare areas where inequity in access across sexual orientation groups is found that are not justified based on need. Mandated LGBTQ+-specific training of the healthcare workforce may help address some barriers of access for these groups. The study aims to understand the potential economic implications for mandated LGBTQ+-specific healthcare training on the healthcare system in England, UK to inform commissioning of training provision. **Methods**: Cervical cancer screening was used as an exemplar case where there appears to be inequity in access for different sexual orientation groups. A decision model was developed and analysed that considered the impacts of greater uptake of screening for lesbian and bisexual women due to LGBTQ+ training. Costs took the perspective of the healthcare system and outcomes modelled were cancer cases averted in a timeframe of 5 years. **Results**: Based on cervical cancer screening alone, where training costs are fully attributed to this service, training would likely result in fewer cancer cases detected in the lesbian and bisexual populations, though this comes at a modest increase in healthcare sector costs, with this increase largely reflecting a greater volume of screens. Training costs do not appear to be a major component of the cost implications. **Conclusions**: In resource-constrained systems with increasing pressures for efficiency savings, the opportunity cost of delivering training is a realistic component of the commissioning decision. The findings in this paper provide a signal that mandated LGBTQ+ training in healthcare could lead to potentially greater outcomes and in breaking down barriers of access and could also enable the healthcare system to provide more equitable access to healthcare.

## 1. Background

Many health systems worldwide aim to provide equitable access to health care [1]. In England, public sector healthcare providers are required to adhere to the Equality Act [2], whereby providers need to ensure equality of opportunity across protected characteristics including age, disability, gender reassignment, pregnancy and maternity, race, religion or belief, sex, and sexual orientation. In addition, the National Health Service (NHS), in its constitution aims to deliver access to healthcare services based on need with an additional requirement not to deliver services based on the ability to pay [3].

Sexual minority orientation can be categorised by attraction (such as same-sex attraction), identity (lesbian, bisexual), sexual behaviour (women who have sex exclusively with women (WSEW), women who have sex with men and women (WSMW)), or by partnership status (same-sex civil partnership or marriage). Under the aim of equitable access, the use of services should be driven by need and not sexual orientation. There is evidence demonstrating that a smaller proportion of lesbians, bisexual women, and women who have sex with women attend for cervical smears than their heterosexual counterparts [4,5]. However, results from the UK NatSAL survey demonstrated the there was a higher rate of heterosexual sex in women who subsequently reported same-sex genital contact compared to those who did not report same-sex genital contact [6]. Subsequently it has been found that there is a higher rate of teenage pregnancy in lesbians and in bisexual women compared to heterosexual women [7]. This suggests that there might be a higher rate of Human Papillomavirus (HPV) and therefore subsequent cervical cancer in lesbians and bisexual women compared to heterosexual women [8]. The fact that there is a need for cervical smears among these population groups but lower rate of attendance of these screenings for lesbian and bisexual women suggests the presence of inequity in access to healthcare services. Inequity in access is a violation of the NHS Constitution and may result in inequities in health. Given this, there is a clear need for the healthcare system to address these access issues.

The cause of inequity in access may be multifaceted. Access is a complex construct, determined by factors such as the availability and awareness of services, the accessibility of services, acceptability of providers and patients, and affordability concerns [9,10]. Poorer rates of cervical smears for lesbian and bisexual women may be explained by several factors. From an awareness and availability aspect, until 2009, government policy was that lesbians were not eligible for cervical smears, as it was assumed that they would be at lower risk of cervical cancer [5], this may explain why some sexual minority women have been refused smears even though they have attended (14% (70/500) [4], 6% (7/114) [11]). The acceptability of both patient and provider of cervical screening may contribute to inequity in access. Qualitative research about why lesbians and bisexual women are less likely to attend cervical screening has identified several factors. These include discomfort with the procedure [4], feeling pressured to disclose their sexual orientation, or facing negative reactions upon disclosure, such as gasping, recoiling, apparent repulsion, or receiving inappropriate lectures from smear nurses regarding heterosexual sexual activity [12].

Addressing inequities in access is the responsibility of the provider, yet few UK health professionals currently receive training in sexual orientation and gender identity (SOGI) health during their undergraduate or postgraduate courses [13]. Apart from a few exceptions, UK courses with lesbian, gay, bisexual, trans, queer, and other minority (LGBTQ+)-specific content tend to be limited in scope and tend to be influenced by individual activist academics within educational establishments who have a specific interest in LGBTQ+ health [14,15]. Indeed, a relatively high proportion of health staff consider that sexual orientation is not relevant to a patient’s health needs [16]. This is also the case for healthcare trainers [13]. However, the research described above plus high-level influential reports, such as the UK Parliament Women and Equalities Select Committee Report into Health and Social Care and LGBT Communities [17], suggest that healthcare professionals need to be much more knowledgeable about LGBT health issues.

Since there have been education and training materials on SOGI health available to NHS staff for a number of years, and Core Training Standards in Sexual Orientation for Health Professionals have been available since 2006 [16], the problem has not been availability of training materials. The barriers are more likely to be unwillingness by healthcare staff to accept the need for this knowledge, and an assumption that this is additional training that would come at a considerable additional cost in terms of time and resources.

With scarce healthcare resources, mandated training, whilst having the potential to help address inequities in access, may require evidence of the cost and impacts to equip commissioners and planners with the economic evidence for which to make the decision for formalising SOGI health training. This paper aims to inform this evidence with an assessment of the impacts and costs of introducing formal education training in SOGI health issues for healthcare providers. Cervical smears are used as an exemplary case for measuring impacts (SOGI health issues will be much broader, so this represents an underestimate of the anticipated benefits).

## 2. Materials and Methods

To assess the cost-effectiveness of mandated LGB training in healthcare providers we developed a decision analytical model using a decision tree approach to evaluate. The study is presented in accordance with CHEERS standards [18] (Appendix A). Data were incorporated from existing sources; hence the study did not require ethical approval.

### 2.1. Target Population and Setting

The target population comprised persons identifying as lesbian or bisexual in the UK population eligible for cervical screening under the current NHS Cervical Screening Programme (NHSCSP). The current UK cervical screening policy states that all women with a cervix between the ages of 25 and 64 years are sent invitations via letter every three years up until the age of 50 and every five years beyond that. The first step of screening is to perform an HPV test; if the results are negative, no further action is required because there is a low chance of cervical cancer (CC) or precancerous abnormalities. A smear test will be performed if the HPV test is positive. If the smear test is negative, they’ll be invited back for another screening in a year and a second time in two years if the HPV infection continues. A colposcopy is advised if HPV positive is still present after three years. A colposcopy is recommended when the HPV test is positive and abnormal cells are seen on the smear test. The goal of the cervical screening programme is to identify cervical abnormalities early, before they develop into CC, and so lower the incidence of CC in the population [19].

We considered different population age groups for sensitivity analysis. Data is taken from the Office of National Statistics (ONS) (Table 1).

Mandated training in LGB healthcare will have wider-reaching impacts than on this population (and potentially additional impacts on this population) than on cervical screening alone. However, we chose cervical screening as an exemplary case of the potential impacts of such training.

### 2.2. Study Perspective

The healthcare provider perspective is used for costing, as the cost of the screening procedures and training would be covered by NHS England. The healthcare provider perspective is in accordance with the National Institute for Health and Care Excellence (NICE) guidelines for economic evaluation [20].

### 2.3. Intervention and Comparator

Mandated training to healthcare professionals about LGB issues is compared to current practice (voluntary uptake of training in relatively few health professionals). A mandated training programme would focus on educating professionals on LGB health issues, aimed at providing appropriate services to LGB population groups. The training would prioritize strategies aimed at fostering a welcoming environment to ensure that this population feel comfortable accessing all required healthcare services [16].

### 2.4. Model Parameters

The decision tree model is conceptualized by following the guidelines set by Roberts et al. [21]. The model considered a 5 years’ time frame. According to NHSCSP, this time frame comprises the maximum amount of time that each woman is eligible for a screening (a complete cycle).

### 2.5. Measure of Effectiveness

Health outcomes are measured using the number of CC cases averted. The resulting economic evaluation is thus a cost-effectiveness evaluation. Alternative outcome measures that span a longer timeframe could include, for example, quality-adjusted life years (QALYs, cost–utility analyses) or mortality. In this specific example our aim is to highlight the removal of barriers in access to healthcare use, for which screening uptake and the direct implications on cancers averted is captured. There is limited evidence around cervical cancer screening in the LGB population. A scoping review that included grey literature was conducted to identify smear uptake rates and potential changes in these following approaches to remove barriers of access.

### 2.6. Measures of Costs

The cost of the screening covers the cost of sample collection, HPV and cytology testing as well as the cost of training healthcare professionals in LGB issues, and the cost of CC treatment. The currency is Pound Sterling (£GBP) at 2022 prices with prices inflated in accordance with inflation estimates [22].

The cost of testing is collected from the literature. The cost of the testing includes consumables, equipment, staff time (sample preparation and reading time), and other laboratory overheads.

The cost of treatment is also collected from the literature. The cost of the treatment is based on observed treatment preferences as a function of the stage of the cancer at diagnosis, including cone biopsy or loop excision, trachelectomy, hysterectomy alone, radiotherapy (with or without hysterectomy), chemotherapy and chemo-radiotherapy (with or without hysterectomy).

The cost of the training is assumed to be between £50 and £100 per health professional. Training would encompass LGB health issues, in particular about how not to deter lesbians and bisexual women from attending for a cervical smear, and include the cost of the educator, education materials and room hire/screen time.

### 2.7. Model Design

Two decision trees are constructed for lesbian and bisexual female populations separately (Figure 1 and Appendix A). The decision tree approach was chosen due to the single event of screening over the timeframe studied (we assume only a single screen would occur in the 5 year timeframe). The parameters of the model reflect the chance of a screening being conducted and subsequent diagnostic outcomes for those that have a screen and those that do not. Diagnostic outcomes for those not having a screen would arise due to investigations following non-screen-related consultations.

**Figure 1 behavsci-14-00260-f001:**
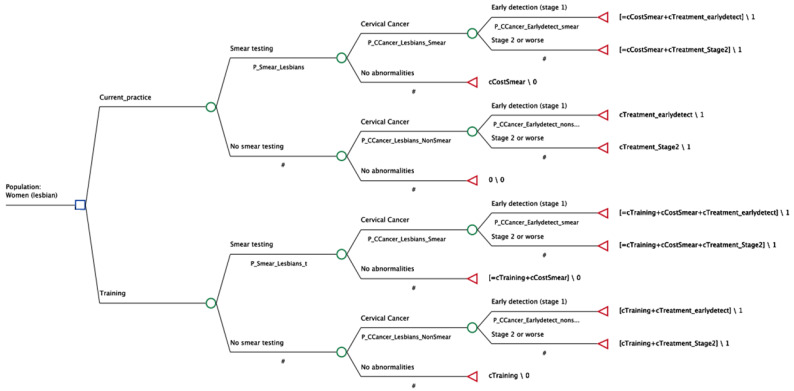
Decision Tree for lesbian female population. P_Smear_Lesbians: Probability of smear uptake in lesbians in standard of care, P_Smear_Lesbians_t: Probability of smear uptake in lesbians with training, P_CCancer_Lesbians_Smear: Probability of CC in lesbians who do smear screening, P_CCancer_Lesbians_NonSmear: Probability of CC in lesbians who do not do smear screening, P_CCancer_Earlydetect_smear: Probability of CC at early stage in population who do smear screening, P_CCancer_Earlydetect_nonsmear: Probability of CC at early stage in population who do not do smear screening. One-way sensitivity analyses were conducted to identify the sensitivity of the model to variations in the cost of delivering training and the patient population impacted by the training. A summary of the key design criteria is included in Table 2. Decision nodes—represented by squares; Chance nodes—represented by circles; End nodes—represented by triangles; #—residual of the branch.

The decision trees are developed to follow the pathway of lesbian and bisexual female population being screened for CC to the final outcome of CC case averted at early stage or stage 2 or worse. Specifically, the models have two arms reflecting mandated or non-mandated training to healthcare professionals. Each arm is subsequently split into two pathways for a regular smear test occurring or not, based on the population prevalence for smear testing. Pathways from the smear tests include CC diagnosis, stratified into early and stage 2+, and no abnormalities. For those without a smear there are similar pathways though diagnosis here that would be outside of routine smear screening.

### 2.8. Sensitivity Analyses

The findings of the model rely heavily on the parameters sourced from the literature and decisions made about the cost of training. To explore the sensitivity of the findings we conduct several one-way sensitivity analyses:Training costs: Where we vary the volume of the population being affected by training.Training effectiveness: Where we vary the effectiveness of the training.Population covered: Where we vary the population age groups affected by training.

## 3. Results

### 3.1. Study Parameters

All the parameters used in the model are provided in Table 3. The probability of smear uptake and CC detection for the lesbian and bisexual populations are taken from the literature [5]. As screenings detect cervical abnormalities that, if untreated, could develop into cancer, the population that gets regular screenings would have a low risk of developing CC. According to the literature, the number of CC diagnoses without screening would be 2.53 times higher [23]. We assume that both population groups would experience similar greater probabilities of CC detection among those who do not have a screen.

Due to the training of healthcare professionals, smear uptake in both population groups would likely increase. The findings of a project by the Lesbian & Gay Foundation (now the LGBT Foundation) and the University of Salford, funded by the NHSCSP, show that after an awareness-raising campaign, the uptake of smear tests among the lesbian population increased by 22% [4]. We assumed that the training provided to healthcare professionals about LGB issues would have an impact of a similar size for the lesbian population and smear uptake in the bisexual population would increase to the level of heterosexuals because of this training (applying a similar 22% rise would result in bisexuals having greater rates of screening, which we felt was a strong assumption to make).

The cost of the training is added to the smear testing cost and the treatment cost at different stages. The latter is added to reflect the impact of more patients diagnosed at early stage in the smear testing branch. The calculation to convert the cost to train one healthcare professional to a per-screen training cost is explained in Figure 2. The training cost for each test is determined by assuming an additional 1, 2, and 3 persons from the population are screened per month (due to training), for a total of 60, 120, 180 additional screens throughout the course of the study (5 years). This results in £1.25, £0.63, and £0.42 per-screen costs, respectively. In the primary analysis, we considered the per-screen cost (£0.63), while sensitivity analysis was performed using the other per-screen costs (£1.25 and £0.42).

Incremental costs and outcomes are provided in Table 4. The model identifies that training healthcare professionals about LGB health issues will increase smear uptake in both population groups which results in an increase in identification of pre-cancer abnormalities and cancer at early stages (179 compared to 147 without mandated training for lesbians and 660 compared to 639 for bisexual women). The rise in the population being screened results in a smaller population not being screened which explains the decline in cancers averted in the not-screened group.

The incremental cost is greater for lesbian women, this is driven by a greater number of screens being conducted which more than offsets the lower costs of treatment in this group. For bisexual women there is a lower incremental cost, this is due to the smaller increase in screening with lower treatment costs now more than offsetting the increase in screening costs.

The incremental cost-effectiveness ratios are £23,467 per cancer averted for lesbians and −£1463 per cancer averted for bisexuals.

### 3.2. Sensitivity Analyses

Uncertainty in the parameters were considered in relation to costs of training, the effectiveness of training, and the population being affected. Cost per cancer averted increases with the cost of training in the lesbian population, though this is fairly minimal (Table 5). For the bisexual female population, lower costs of training impacts whether mandated training dominates standard care (Table 5).

The effectiveness of training is not known. We assume changes based on a previous study aiming to improve awareness [4]. This study targeted the population of patients rather than healthcare professionals but indicates how amenable the population may be to change. Given the lack of applicability of this study to mandated training to healthcare professionals, we conducted sensitivity analyses that varied the effectiveness of mandated training from zero (no effect) to an effect that results in the same screening uptake as heterosexuals (a 22 percentage point increase). For bisexual women, given screening uptake is greater than amongst lesbian women, the sensitivity analyses ranged from zero (no effect) to 4 percentage points.

Where the training has no effect, the mandated training would cost £150,570 and £122,850 for lesbian and bisexual women, respectively. For lesbian women, increasing effectiveness reduces the ICER with negligible reductions in the ICER from around 6–7 percentage points greater uptake (Figure 3). For bisexual women, training becomes a dominant strategy should it increase uptake by 3 percentage points and has a low ICER with small effects on uptake (1–2 percentage points, Figure 4).

Varying the population covered does not impact on the ICER, this is because the parameters in the model are not sensitive to the population covered. Figure 5 and Figure 6 present the expected costs and outcomes for various population groups for lesbians and bisexual women, respectively. Widening the population results in greater expected costs and greater number of cancers detected.

## 4. Discussion

The project sought to understand the potential economic implications of mandated LGB healthcare training. Cervical screening was used as an exemplar of the impacts. Based on this example alone, where training costs are fully attributed to screening activity, we find training would result in fewer cancer cases in the lesbian and bisexual female populations though this comes at a modest increase in costs to the healthcare sector, with this increase largely reflecting a greater volume of screens.

Our aim was to provide an indication of the impacts of mandated training. There are numerous limitations of the approach taken that broadly reflect the complexity of modelling screening programmes. Our approach was not one to inform the screening evidence base but to highlight the potential impacts mandated training in LGB healthcare could have. There are a number of ways the analytical approach could be strengthened. These include apportioning costs of training to the broader LGB population rather than attributing them to lesbians or bisexual women alone (in the model here we assume training impacts only cervical screening or is cervical screening-specific), this would however have minimal impact on the findings here beyond further strengthening the dominance of mandated training in the bisexual female population.

Our analyses assessed cancers detected (and hence averted) as the outcome measure of effectiveness, more comprehensive approaches would take a longer timeframe of the impacts on outcomes and incorporate impacts on QALYs and/or mortality—particularly relevant given our costs include the costs of treatment; to this extent our analyses overstate the costs for cancer detection and aversion. In a model taking a longer timeframe and an outcome perspective incorporating quality of life (which would be anticipated to find positive incremental outcomes due to cancer detection), our analyses provide a strong signal that this would likely be considered cost-effective under standard cost-per QALY thresholds used by the UK healthcare system.

Our analysis was limited by the current evidence base concerning cancer prevalence and detection in lesbians and bisexual women, particularly with regards to age groupings where we assume cancer prevalence is uniform. Future research could develop the model used here to incorporate various rates of prevalence of cancer in lesbian and bisexual women of different age groups once evidence is available.

Although there is literature about the cost-effectiveness of cervical screening [24], there has not been a focus on professional training to remove barriers in access. We assume a rate of increase in screening uptake for lesbians and bisexual women that was informed by a limited evidence base. As further research evolves, highlighting the impacts of interventions to improve access for these groups, the model could be adapted to show the sensitivity of this assumed rate of change.

Our analyses do not include the cost of any screening activities undertaken for the identification of cancer in the non-mandated training arm. There will likely be costs associated with the identification of cancer outside of screening services meaning the costs in the non-mandated arm will underestimate the costs here.

Our analyses assumed non-recurrence in a 5 year time frame and did not consider a longer timeframe of screening (e.g., where individuals may have multiple screens over their lifetime).

The analyses use training as an exemplar with limited discussion about the components of this or the frequency at which this may be conducted. We also assume training would only impact the 5 year time frame. With a limited evidence base, we conducted sensitivity analyses varying the effectiveness of the training. The results suggest that training approaches that improve uptake of only a few percentage points may be considered cost-effective. Evidence on the effectiveness of mandated training on uptake is needed to inform how applicable/likely this is. Our approach, over a longer time frame may be adapted to include training at each 5 year time frame, rather than a single event.

Our approach used screening as an exemplar, this was a particularly problematic example given screening comes at a cost that may or may not provide an outcome benefit for everyone. To this extent, similar approaches to those taken here but focused on specific treatments may find mandated training to result in lower costs of healthcare borne due to more timely access to treatments and treatment adherence. Future research could explore this assumption.

In a system where access to healthcare should be based on need, the decision-making criterion for mandated training should not solely be based on arguments of cost-effectiveness. However, in a resource-constrained system with increasing pressures for efficiency savings and competing innovative ways to deliver care, the opportunity cost of delivering mandated training is a realistic determinant. The findings in this paper provide a signal that mandated LGB training in healthcare could lead to potentially greater outcomes and in breaking down barriers of access and could also enable the healthcare system to provide more equitable access to healthcare. Whilst this may result in greater costs for the healthcare system in screening programmes, the impacts on other healthcare services may result in net savings.

## 5. Conclusions

Barriers to accessing healthcare for LGB populations have the potential to widen or sustain inequalities in health. Training health professionals to deliver more accessible healthcare is one way to try to break down these barriers. However, in resource-constrained healthcare systems facing mounting pressures for efficiency savings, the potential opportunity cost of investing in training must be considered in commissioning decisions. The findings presented in this paper suggest that mandating LGBTQ+ training for healthcare professionals could not only improve equity in access but also generate efficiency gains.

## Figures and Tables

**Figure 2 behavsci-14-00260-f002:**
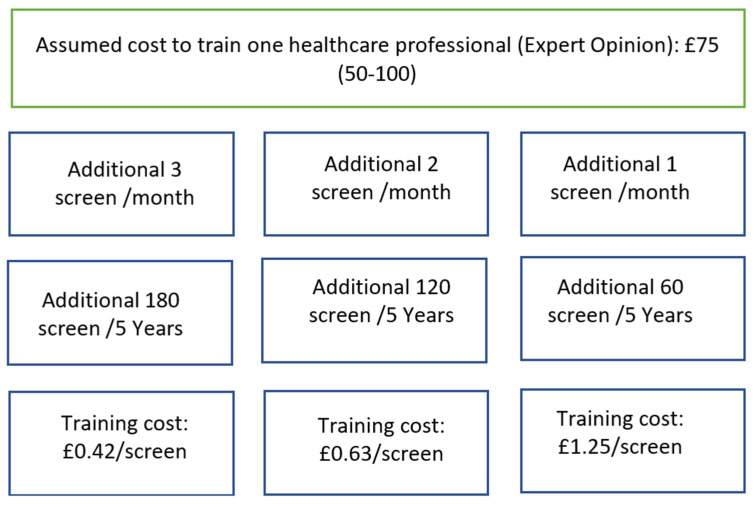
The calculations of per-screen training cost. Note: Additional means the increase in the number of lesbians or bisexual women who received a smear due to training of the health professional.

**Figure 3 behavsci-14-00260-f003:**
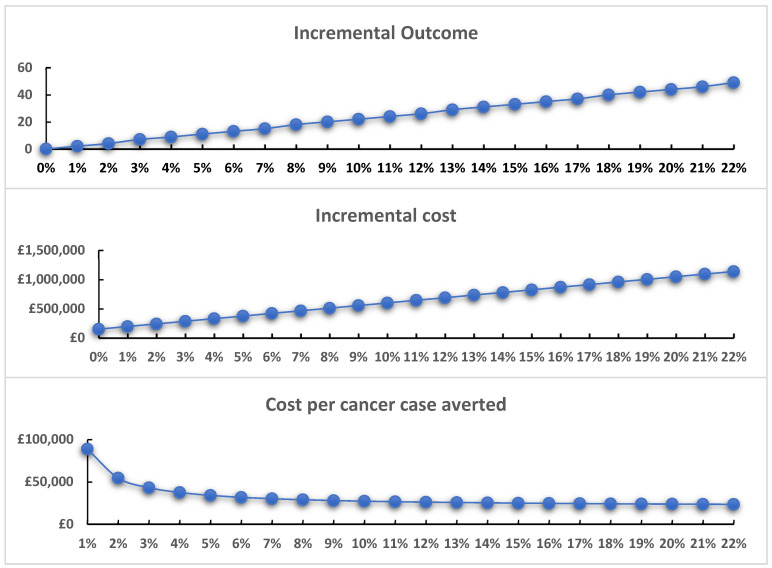
Mandated training effectiveness uncertainty: lesbian population.

**Figure 4 behavsci-14-00260-f004:**
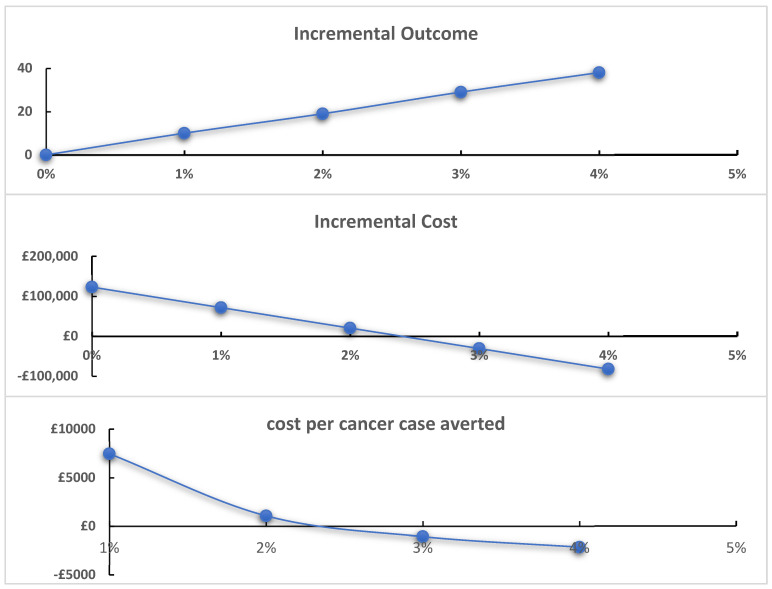
Mandated training effectiveness uncertainty: bisexual female population.

**Figure 5 behavsci-14-00260-f005:**
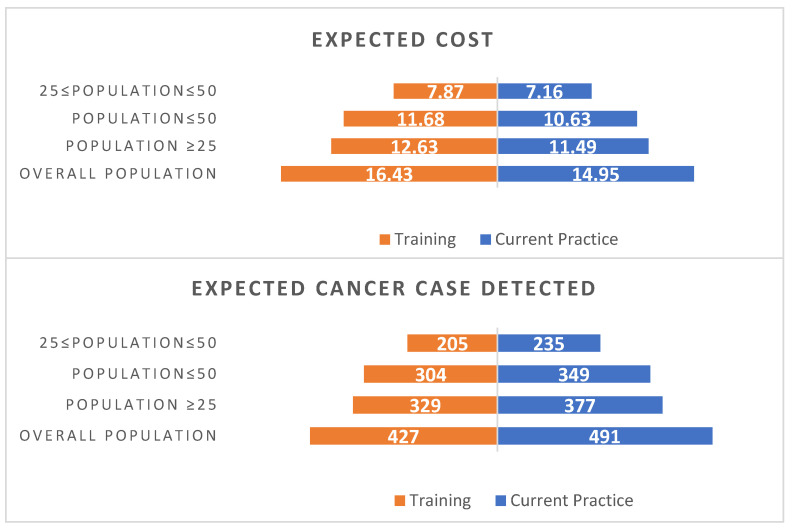
Different population age groups for lesbian population, UK. Note: The expected cost is in million pounds Sterling and assumed training cost per screen is £0.63.

**Figure 6 behavsci-14-00260-f006:**
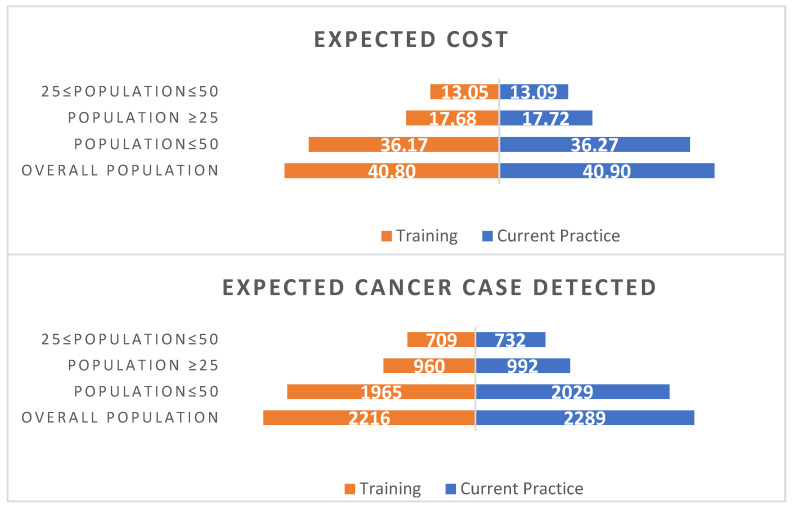
Different population age groups for bisexual female population, UK. Note: The expected cost is in million pounds Sterling and assumed training cost per screen is £0.63.

**Table 1 behavsci-14-00260-t001:** The Distribution of Lesbian and bisexual population (in thousands), UK.

Sexual Identity	Overall Population	25 ≤ Population ≤ 50	Population ≤ 50	Population ≥ 25
Lesbian	311	149	221	239
Bisexual	450	144	399	195
Heterosexual or straight	25,680	10,170	13,059	22,791

Source: ONS UK, 2020.

**Table 2 behavsci-14-00260-t002:** Key design criteria.

Decision problem	What are the benefits and costs of introducing formal education training in LGB health issues for healthcare providers? Cervical smears are used as an exemplar case for measuring impacts
Intervention	Mandated training to healthcare professional about LGB issues
Comparator	The comparator is the current standard of care in the United Kingdom
Population	Population groups of persons identifying as lesbian and bisexual in UK
Model type	Decision Tree
Software	TreeAge and Excel
Time horizon	5 years’ time frame (this time frame comprises the maximum amount of time that each person is eligible for screening)
Study perspective	National Health Service (NHS) (health system perspective)
Cost	National currency (£) at 2022 prices *
Consequences	Cancer case averted
Uncertainty	Deterministic: one-way sensitivity analysis/scenario analysis

* Unit costs were inflated to 2022 prices.

**Table 3 behavsci-14-00260-t003:** Parameters for the model (Population: lesbian and bisexual).

Parameters	Current Practice	Training	Source
Probability of smear uptake in lesbians	61.50%	75.03% *	[5]
Probability of CC in lesbians who do smear screening	0.10%	0.10%	[5]
Probability of CC in lesbians who do not do smear screening	0.25%	0.25%	[23]
Probability of smear uptake in bisexuals	81.90%	84.6% **	[5]
Probability of CC in bisexuals who do smear screening	0.40%	0.40%	[5]
Probability of CC in bisexuals who do not do smear screening	1.01%	1.01%	[23]
Probability of CC at early stage in population who do smear screening	83.5%	83.5%	[23]
Probability of CC at early stage in population who do not do smear screening	64.8%	64.8%	[23]
**Cost of screening *****			[22]
Sample collection	£18.22 (14.88, 22.17)		
HPV test per sample *	£11.60 (8.61, 15.51)		
Cytology test per slide *	£21.60 (17.79, 26.21)		
Total	£51.43		
**Cost of training**	£ (50–100) ****		Assumption
**Cost of treatment**			[22]
Stage I	£5498 (4886, 6137)		
Stage II	£24,642 (21,337, 27,981)		
Stage III	£24,265 (20,993, 27,981)		
Stage IV	£20,614 (17,798, 23,814)		
Average cost of stage II, III, and IV	£23,174 (16,253, 21,478)		

* Due to training, the smear uptake in the lesbian population is increased by 22%. ** Due to training, we assumed that the smear uptake in the bisexual female population will be equal to heterosexual women. *** Cost of one smear test performed (in Pounds Sterling) and inflated to 2022. **** Cost of LGB ** training to healthcare staff (half-day training).

**Table 4 behavsci-14-00260-t004:** Results: Cost per cancer case averted for population ≥ 25, UK.

	Current Practice		Training	
Lesbian Population				
	Expected Cost (£m)	Expected Outcome (Cancer Detected)	Expected Cost (£m)	Expected Outcome (Cancer Detected)
Screened	8.80	147	10.84	179
Not screened	2.70	230	1.79	149
Total	11.49	377	12.63	329
Incremental cost			1.14	
Incremental outcome (cancers detected)				−49
Incremental outcome (cancers averted) *				49
Cost per case averted			23,467	
**Bisexual Female Population**				
Screened	13.59	639	14.14	660
Not screened	4.14	353	3.54	300
Total	17.72	992	17.68	960
Incremental cost			−0.05	
Incremental outcome (cancers detected)				−32
Incremental outcome (cancers averted) *				32
Cost per case averted			−1463	

Note: At a training cost per screen of £0.63; * cancers averted are the difference between cancer cases detected in both the arms (the mandated training arm and not mandated training arm).

**Table 5 behavsci-14-00260-t005:** Training cost uncertainty: Cost per cancer case averted for population ≥ 25, UK with varying training costs.

Training Cost	Current Practice			Training			Incremental Cost	Incremental Outcome (Case Detected)	Incremental Outcome (Case Averted)	ICER
	Screened	Not Screened	Total	Screened	Not Screened	Total				
**Lesbian Population**										
0.42	8.796	2.696	11.492	10.806	1.774	12.580	1.088	−49	49	£22,432
0.63	8.796	2.696	11.492	10.844	1.786	12.630	1.138	−49	49	£23,467
1.25	8.796	2.696	11.492	10.955	1.823	12.778	1.286	−49	49	£26,522
**Bisexual Population**										
0.42	13.588	4.136	17.725	14.106	3.532	17.638	−0.087	−32	32	−£2759
0.63	13.588	4.136	17.725	14.140	3.538	17.679	−0.046	−32	32	−£1463
1.25	13.588	4.136	17.725	14.243	3.557	17.800	0.075	−32	32	£2364

Note: The expected cost is in million pounds.

## Data Availability

All data generated or analysed during this study are included in this published article [and its Appendix A] and is also available from the corresponding author on request. We collected the population estimates from the Office of National Statistics and data on different parameters from published literature (Saunders et al., 2021 (DOI: 10.1177/0969141320987271) [5]; Landy et al., 2016 (DOI: 10.1038/bjc.2016.290) [23]; Bains et al., 2019 (DOI: 10.1136/ijgc-2018-000161) [22]).

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
