# Peer review of "What Are the Economic Arguments for Mandating LGBT+ Health Training for Healthcare Providers? An Economic Evaluation of the Impacts of LGBT+ Health Training on Cervical Screening"

_behavsci, 2024, doi:10.3390/bs14030260_

Round 1

Reviewer 1 Report

Comments and Suggestions for Authors

I appreciate the opportunity to review this manuscript on an important topic.  Given the growing rates of both LGBTQ+ cancer survivors and the cancer inequities they face, the manuscript addresses a critical exploration of the financial impact and feasibility of LGBTQ+ provider trainings for reducing cancer disparities. My comments are intended to strengthen the project.

I am not clear what is meant on page 2 line 48 with “UK NatSAL survey demonstrated the there was a higher 47 rate of heterosexual sex in women who subsequently reported same sex genital contact 48 compared to those who did not report same sex genital contact.”  Could you more explicitly state how heterosexual sex and pregnancy relate to rates of cervical cancer?

Please edit for clarity the wording on page 2 line 64 in the sentence starting with “acceptability of both…”

Should page 2 line 89 be “healthcare” rather than health and care resources?

It would be helpful to strengthen the rationale for focusing on cervical cancer screenings.  Please make explicit how screening rates relate to diagnosis rates, including stage of diagnosis, and to treatment outcomes.

I am struggling to follow the conclusion on page 8, line 246 that the incremental cost is greater for lesbian women.  Please explain the manuscript how that incremental cost was calculated and why it differs so greatly for lesbian and bisexual women.  There is also a grammatical error in that paragraph.

On page 9, line 271, should that be “effect” rather than “affect”?

Comments on the Quality of English Language

None

Author Response

Point to point responses are added in the attached file and pasted below.

Review Report (Reviewer 1)

I appreciate the opportunity to review this manuscript on an important topic.

Given the growing rates of both LGBTQ+ cancer survivors and the cancer inequities they face, the manuscript addresses a critical exploration of the financial impact and feasibility of LGBTQ+ provider trainings for reducing cancer disparities. My comments are intended to strengthen the project.

  1. I am not clear what is meant on page 2 line 48 with “UK NatSAL survey demonstrated the there was a higher 47 rate of heterosexual sex in women who subsequently reported same sex genital contact 48 compared to those who did not report same sex genital contact.” Could you more explicitly state how heterosexual sex and pregnancy relate to rates of cervical cancer?

Response: we have added further clarity on page 2, line 52.

  1. Please edit for clarity the wording on page 2 line 64 in the sentence starting with “acceptability of both…”

Response: The comment has been incorporated (please see page 2 lines 71-78)

  1. Should page 2 line 89 be “healthcare” rather than health and care resources?

Response: The comment has been incorporated (please see page 2 lines 103)

  1. It would be helpful to strengthen the rationale for focusing on cervical cancer screenings.

Response: The comment has been incorporated (please see page 2 lines 56-61)

  1. Please make explicit how screening rates relate to diagnosis rates, including stage of diagnosis, and to treatment outcomes.

Response: Please note the manuscript provides a general discussion starting from first step of screening which is a HPV test to early diagnosis of abnormalities before they develop into Cervical Cancer, this is explained in section 2.1 (Target population and setting). Specific context to our example is explained in section 2.7 (Model design) where the whole pathway of lesbian and bisexual female population being screened for CC to the final outcome of CC case averted at early stage or stage 2 or worse is explained, with the specific rates provided in section 3.1.

  1. I am struggling to follow the conclusion on page 8, line 246 that the incremental cost is greater for lesbian women. Please explain the manuscript how that incremental cost was calculated and why it differs so greatly for lesbian and bisexual women.

Response: Please note we explain this in the manuscript (lines 268-272): The incremental cost is greater for lesbian women; this is driven by a greater number of screens being conducted which more than offsets the lower costs of treatment in this group. For bisexual women there is a lower incremental cost, this is due to the smaller increase in screening with lower treatment costs now more than offsetting the increase in screening costs.

  1. There is also a grammatical error in that paragraph. On page 9, line 271, should that be “effect” rather than “affect”?

Response: The comment has been incorporated (please see page 10 lines 293)

Reviewer 2 Report

Comments and Suggestions for Authors

The aims of this research were to understand the potential economic implications for mandated LGBTQ+-specific healthcare training on the healthcare system in England, UK to inform commissioning of training provision. This article has significance in various context. It addresses the health problems of lesbian, gay, bisexual, trans, queer and other minority population, an area rarely researched. It is very interesting, relevant, and contributing to the existing knowledge in the field.

Although the model adopted in this research is heavily dependent on the parameters sourced from the existing literature including decisions on the costs components; however, the authors succeeded in achieving their objectives.

The article is well conceived and executes. Introduction section clearly brings about the justification and contribution of this research. The methodology section is adequately presented as the study followed CHEERS checklist. Data analysis and presentation of results with tables and figures are found to be adequate. References used in the manuscript seems to be appropriate.

However, there are few comments which need to be considered.

1.     In materials and method section, sub section 2.3 Intervention and Comparator need more elaboration, especially more details about the “intervention” ie., mandated training to healthcare professional about LGB issues is required.

2.     Line 146-148- Please give the source for “A scoping review that included grey literature was conducted to identify smear uptake rates and potential changes in these following approaches to remove barriers of access”.

3.     Line 162- “The cost of the training is assumed to be between £50 and £100 per health professional training’-please give the source / on what basis this is calculated or assumed.?

4.     Line 297-298 “The project sought to understand the potential economic impacts of mandated LGB healthcare training” - the sentence is misleading and it is better to use “economic implications” rather than “economic impact”, which is a wider concept. This study measured costs from providers’ perspective only.

5.     Discussion section may be elaborative by corroborating the findings from other CEA studies on health professional training on cervical screening of other population, if possible.

6.     Conclusion section is missing in this manuscript

7.     Authors may like to include any recent literature on this issue, if available

Author Response

Point to point responses are added in the attached file and also pasted below.

Review Report (Reviewer 2)

The aims of this research were to understand the potential economic implications for mandated LGBTQ+-specific healthcare training on the healthcare system in England, UK to inform commissioning of training provision. This article has significance in various context. It addresses the health problems of lesbian, gay, bisexual, trans, queer and other minority population, an area rarely researched. It is very interesting, relevant, and contributing to the existing knowledge in the field.

Although the model adopted in this research is heavily dependent on the parameters sourced from the existing literature including decisions on the costs components; however, the authors succeeded in achieving their objectives.

The article is well conceived and executes. Introduction section clearly brings about the justification and contribution of this research. The methodology section is adequately presented as the study followed CHEERS checklist. Data analysis and presentation of results with tables and figures are found to be adequate. References used in the manuscript seems to be appropriate.

However, there are few comments which need to be considered.

  1. In materials and method section, sub section 2.3 Intervention and Comparator need more elaboration, especially more details about the “intervention” i.e., mandated training to healthcare professional about LGB issues is required.

Response: We have expanded our description of what mandated training may include (page 4, lines 148-152) and provided a reference that highlights the needs here.

  1. Line 146-148- Please give the source for “A scoping review that included grey literature was conducted to identify smear uptake rates and potential changes in these following approaches to remove barriers of access”.

Response: This was not a published scoping review, but part of the research conducted for this study for the purposes of identifying metrics to populate the model.

  1. Line 162- “The cost of the training is assumed to be between £50and £100 per health professional training’-please give the source/ on what basis this is calculated or assumed.?

Response: This assumption was made after consulting with experts in the field.

  1. Line 297-298 “The project sought to understand the potential economic impacts of mandated LGB healthcare training” – the sentence is misleading, and it is better to use “economic implications” rather than “economic impact”, which is a wider concept. This study measured costs from providers’ perspective only.

Response: The comment has been incorporated (please see page 12 line 319)

  1. Discussion section may be elaborative by corroborating the findings from other CEA studies on health professional training on cervical screening of other population, if possible.

Response: Although there is a literature around cost-effectiveness of cervical screening [Sun, L., et al. (2023)], there has not been a focus on professional training to remove barriers in access.

  1. Conclusion section is missing in this manuscript.

Response: conclusion section is added (please see page 14, lines 388-395)

  1. Authors may like to include any recent literature on this issue, is available.

Response: Literature addressing this issue is limited, and we have tried our best to include all relevant available literature in this work.
